evolution, ecology, behaviour

cultural evolution, behavioural ecology, biological institutions, evolution of cooperation, numerical simulation

**Author for correspondence:**
Seth Frey
e-mail: sethfrey@ucdavis.edu

# A dynamic over games drives selfish agents to win–win outcomes

Seth Frey[1] and Curtis Atkisson[2]

[1]Department of Communication, and [2]Department of Anthropology, University of California Davis, One Shields Avenue, Davis, CA 95616, USA

 SF, 0000-0002-5298-5089; CA, 0000-0003-3575-6871

Understanding human institutions, animal cultures and other social systems requires flexible formalisms that describe how their members change them from within. We introduce a framework for modelling how agents change the games they participate in. We contrast this between-game 'institutional evolution' with the more familiar within-game 'behavioural evolution'. We model institutional change by following small numbers of persistent agents as they select and play a changing series of games. Starting from an initial game, a group of agents trace trajectories through game space by navigating to increasingly preferable games until they converge on 'attractor' games. Agents use their 'institutional preferences' for game features (such as stability, fairness and efficiency) to choose between neighbouring games. We use this framework to pose a pressing question: what kinds of games does institutional evolution select for; what is in the attractors? After computing institutional change trajectories over the two-player space, we find that attractors have disproportionately fair outcomes, even though the agents who produce them are strictly self-interested and indifferent to fairness. This seems to occur because game fairness co-occurs with the self-serving features these agents do actually prefer. We thus present institutional evolution as a mechanism for encouraging the spontaneous emergence of cooperation among small groups of inherently selfish agents, without space, reputation, repetition, or other more familiar mechanisms. Game space trajectories provide a flexible, testable formalism for modelling the interdependencies of behavioural and institutional evolutionary processes, as well as a mechanism for the evolution of cooperation.

## 1. Introduction

Evolutionary game theory has proved valuable for analysing cooperation in a wide variety of biological and social systems. However, these systems are typically treated as fixed, making it difficult to model the incremental influence agents often have over the incentive structures they face. Whether through evolutionary processes [1,2] or explicit institutional design [3], agents are able to adjust the games they participate in and enact preferred incentive structures. We extend the well-established theoretical tradition that conceives of political, legal, economic, social and 'biological' institutions as a series of economic games [4–7], by allowing agents to choose the next game in a series from among those that 'neighbour' the current one. Linking within-game behaviour and between-game preferences allows us to study emergent complexity in settings ranging from behavioural ecology to governance institutions.

Throughout the animal world, formalisms for evolving games have provided parsimonious models of the intricacies of sexual selection [2], interactions with resource systems [8] and the emergence of diversity [1]. For the human world in particular, incremental changes to game structures offer a rich model of institutional change at the human level, with findings on preferences for punishment [9], fairness/efficiency trade-offs [10], negotiation processes [11] and policy development [3,12].

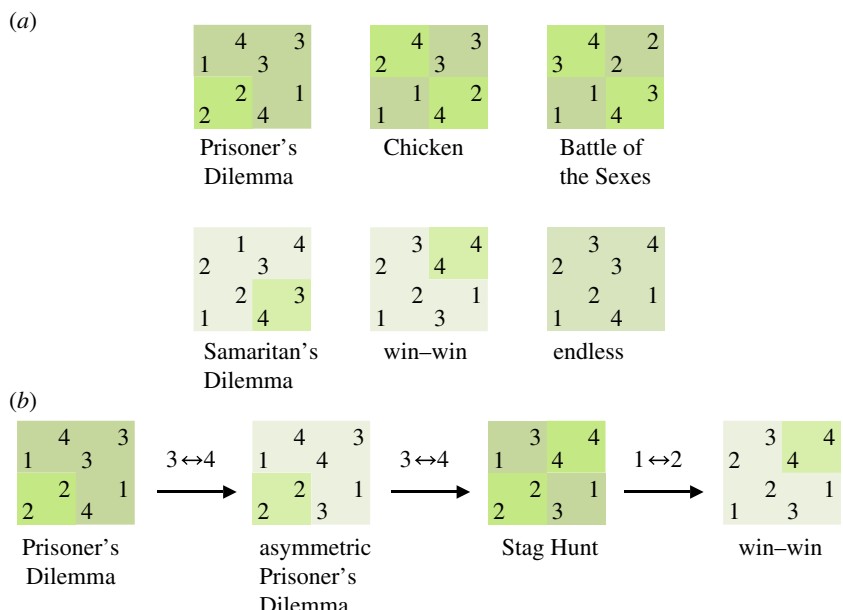

**Figure 1.** A sample of ordinal games, along with a trajectory through them. (*a*) The space of two-player, two-choice ordinal games includes a variety of interesting and relevant games, with well-studied games such as the Prisoner's Dilemma, Chicken and Battle of the Sexes, as well as less remarked upon win–win games that do not require strategy, cyclic games without pure-strategy equilibria and asymmetric games. Ordinal games are defined as games having consecutive integer pay-offs up to the number of outcomes. In the context of game theory, ordinal pay-offs are easily interpretable as indicating a player's ranking of possible game outcomes. In these illustrations, 4 is high and 1 is low. Nash outcomes of a game in the figure are indicated with a slightly brighter green. (*b*) Assuming that institutional change is incremental rather than revolutionary, institutional evolution can be modelled as a trajectory through neighbouring games. In contrast to familiar models describing populations of players, games in this model are played in sequence by the same small group of individuals. Two games in this space are neighbours if they are identical except for two pay-off values being swapped. Here, we illustrate how a pair of agents might incrementally evolve a Prisoner's Dilemma into a win–win game. This trajectory terminates on the game 'win–win', which is an attractor for self-interested agents who prefer stable, predictable and efficient games (defined herein as offering a unique pure-strategy Nash equilibrium that confers a maximum pay-off to the focal agent). Example adapted from Bruns [39]. (Online version in colour.)

Here we present a framework for studying the interactions of within-game 'behavioural evolution'—the familiar purview of game theory—with between-game 'institutional evolution'. Rather than model fluctuating compositions of populations in closed systems, as in evolutionary game theory, we pose a small number of specific agents playing and altering a series of games. The classic game-theoretic approach to institutions understands them as patterns of behaviours, usually equilibria, that emerge within a game [5,13]. By defining institutions as behaviours, it captures the idea that one 'environment' (a game) can support multiple 'institutions' (stable sets of strategies). Under this flexible framework, many tools of game theory have been brought to bear on theories of institutions, such as recent work using multi-level selection to explain the internalization of social norms [14]. However, in equating institutions and behaviour, this approach to institutional analysis struggles to conceptualize the effects of institutions on behaviour.

A more recent framework defines institutions not as behaviours but as games—North's classic 'rules of the game' [4]—and the surrounding environment not as a game but as the space of potential games. Research in this tradition often uses game spaces to catalogue the variety of within-game dynamics [15–19]. These theoretical and experimental studies parallel an increasing amount of observational work based upon comparisons across large numbers of social systems [20–23]. Other studies represent institutions as fixed networks of games [6,24–26]. A framework introduced by Hurwicz, for example, chains several games together to model an institution's development along with its outcomes [3]. In another approach to cross-game behavioural evolution, games are

arranged in a series to induce learning experiences that steer agents towards specific outcomes [27].

Fortunately, researchers are beginning to focus on internally driven institutional change. Working within a bargaining framework, a community of economists in policy analysis are modelling the general scenario in which a group of agents negotiate over a policy lever, play a game parameterized by that lever, update their preferences in response to that game, and iteratively circle back to negotiate under their new preferences [12,28,29]. Similarly, Powers & Lehman combine behavioural types, spatial variation and a parametrized institution space to understand the transition of human civilizations to large-scale populations [30].

Treating normal-form games as toy institutions, we represent institutional change dynamics in terms of trajectories through game space. In our framework, games undergo selection according to players' institutional preferences—the abstract values and qualities they look for in a social system. Institutional preferences fit within a broader academic interest in human preferences over games, culture, norms and language [31–33]. Institutional preferences have attracted specific interest with theories such as Binmore's, that the processes of cultural evolution select for institutions with the features of stability, efficiency and fairness, in that order [34]. Illustrating the potential for applications to policy, researchers have also elicited communities' preferences for the features of local resource management systems [35,36].

Among attempts to explore large game spaces, one space in particular, the Topology of Games, has attracted broad formal attempts to taxonomize or otherwise compare behaviour across a range of games [37,38] (see figure 1*a* for a sample of the variety of games in the space; for two-dimensional

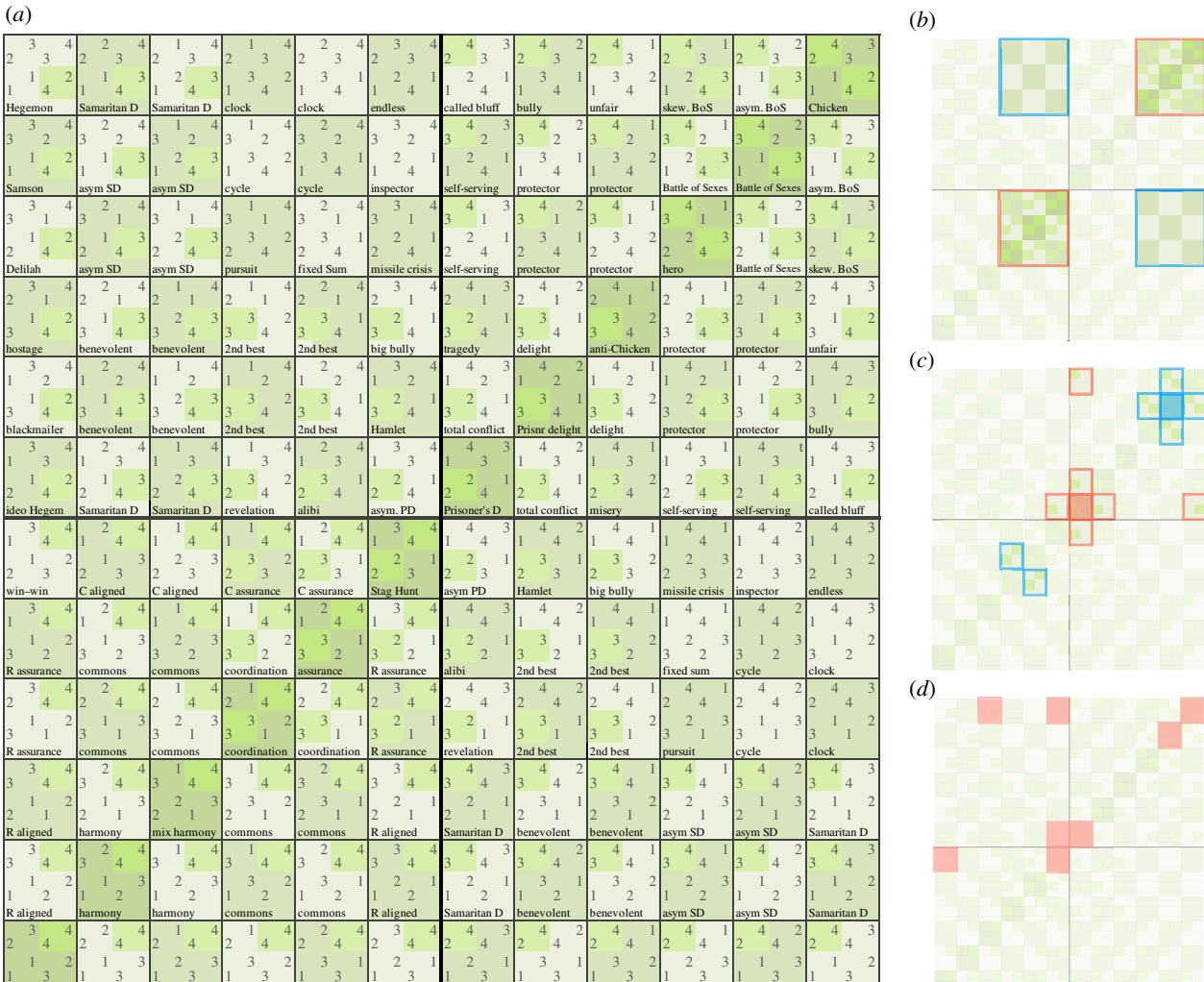

**Figure 2.** The space of two-player games, with masks illustrating its properties. (*a*) A simple representation of the space of 144 two-player, two-choice games with ordinally ranked pay-offs. Observe that symmetric games (electronic supplementary material, figure S2b), which occupy the increasing diagonal, are a minority, and that the lower-left quarter of games are win–win, meaning they have an outcome that confers the maximum pay-off of 4 to both players (also see figure 3*a*). Game spaces beyond two players are much larger and more difficult to diagram than the two-player space, but we present *n*-player results for up to the nine-player $2 \times 2 \times \ldots \times 2$ games. (*b*) This mask of panel (*a*) illustrates the Nash properties of the games in this space. The games in the blue outlines have zero pure-strategy Nash equilibria, while the games in red outlines have two. The remaining games, three-quarters, have exactly one pure-strategy Nash equilibrium. We discuss how this distribution changes as the number of players increases. (*c*) This mask of panel (*a*) illustrates the complex nature of neighbour relations in the two-player space. The red outlines show the six 'neighbours' of the Prisoner's Dilemma. The blue outlines show the neighbours of the Battle of the Sexes. Note that a portion of adjacent games are not visually adjacent, a shortcoming of the two-dimensional grid representation of what is, in fact, a much more complex toroidal topology. (*d*) This mask of panel (*a*) indicates the locations of the games in figure 1*a,b*. (Online version in colour.)

representations of the smallest, two-player, version of that space; see figure 2*a* and the electronic supplementary material, figure S1). The space was first organized as a taxonomy by Rapoport [40]. Its simplicity and structure make it an ideal substrate for modelling the processes of institutional evolution (figure 1*b*) [39].

## 2. Institutional evolution

We introduce a framework for modelling institutional change as a trajectory through an abstract institution space. Specifying a model within this framework involves defining a space, distances over the space, the within- and between-game behavioural rules for agents in the space, and a scheme for selecting the next game from among those preferred by each player. The specific model we define within this framework demonstrates its general potential by introducing within-game-rational agents who traverse a lattice of economic games linked by similarity. They do so in a

hill-climbing process that optimizes for desirable game features such as stability, efficiency and predictability.

## 3. The institutional evolution framework and a model within it

### (a) Institution space

In the most general terms, we understand institutional evolution as a process in which a fixed set of *n* agents play an *n*-player game and then transform it according to their preferences to a neighbouring game, in an iterative cycle (figure 1*b*). The first challenge in making this picture concrete is to find a space of social systems that is rich enough to capture a wide range of human exchange patterns, but simple enough to remain tractable.

We begin with the Topology of Games [37], a space of social systems defined in terms of the possible two-choice

normal-form games with ordinal (ranked) outcomes. Further restricted to the two-player games, this space arranges the 144 games (the 144 unique ways that two agents can assign their own strict rankings over four outcomes [37]) into a highly structured network (figure 2a and electronic supplementary material, figure S1). The Topology of Games has several attractive properties: it is simple, composed of the most elementary class of economic game, and amenable to counting. It is also rich; games within the space represent a broad array of social situations (figures 1a and 2d). The two-player space includes many of the most famous economic games, such as the Prisoner's Dilemma and Chicken, as well as less recognized games, such as no-conflict and win–win games in which individuals' choices lead 'non-strategically' to outcomes that benefit all [41]. As mundane as these 'non-game' games are, their value is clear in the fact that most of our daily social exchanges, themselves the result of evolutionary processes, are similarly mundane [42]. Overall, the space parsimoniously captures an impressive variety of interdependence patterns found in human interactions [40,41], and successfully builds upon the legacy of the most classic and simple games to model institutional processes and behaviour.

A major source of the appeal of the space is its amenability to combinatorics and counting. In general, as population size (number of players) $n$ increases, the number of two-choice ordinal games grows quickly as [37],

$$\frac{(2^n!)^n}{2^n}.$$

This power of a factorial of a power, an astronomical figure, is the number of ways of assigning a player's rankings over a game's $2^n$ outcomes, independently for all $n$ players, with the value in the denominator controlling for double-counting owing to symmetries (electronic supplementary material, figure S2). Although the number of two-player games is in the order $10^2$, the number of four-player games is well above the order $10^{50}$, and the number of eight-player games is much larger than $10^{4000}$. Fortunately, these vast sums do not appear to undermine the countability of the domain.

The Nash equilibria of the Topology's games are more difficult to count directly, but are approachable numerically if not analytically. By direct enumeration, three-quarters of the two-player games have exactly one pure-strategy Nash equilibrium, one-eighth have zero and one-eighth have two (figure 2b) [43]. As population size increases beyond 2, and the number of games grows super-exponentially, the number of expected Nash equilibria per game grows more modestly, approaching a standard Poisson distribution, with a game's chances of having 0 and 1 equilibria equal to $e^{-1}\sim37\%$ and the remaining quarter having 2 or more equilibria [44,45].

The structure of the Topology of Games space is based on the pattern of linkages between neighbouring games (figure 2c). By a conventional definition, two games are neighbours if they differ minimally in pay-offs, meaning they are identical but for the swapping of a 1 and a 2 pay-off, a 2 and a 3, or a 3 and a 4. By this definition, the Stag Hunt neighbours the win–win game because swapping the locations of one player's 1 pay-off and their 2 pay-off turns the former game into the latter (figure 1b shows one possible trajectory from the Prisoner's Dilemma to a win–win game via the Stag Hunt).

Like the number of games, the number of neighbours per game also diverges, implying an even larger explosion in the number of shortest paths between pairs of games, such that simple local strategies such as hill climbing can start to find global optima reliably [46].

## (b) Distance

With the set of games defined, it is possible to introduce a simple conception of distance within the space. We start by restricting our attention to incremental institutional change: trajectories occur over neighbouring games—again, those that differ by just one transposition of similarly ranked pay-offs. The distance between two games is the minimum number of swaps necessary to make them identical.

## (c) A dynamic over game space

Given a space and metric on the space, we can begin to specify dynamics. Agents alternate between playing the current game and choosing which neighbouring game to evolve to. A dynamic in this framework is thus specified in three parts: the definition of player behaviour within a game, the definition of a player's 'institutional preferences' between a game and its neighbours, and the rules for aggregating all agents' neighbour preferences into a single choice. An institutional change trajectory is produced by repeatedly cycling through the steps of playing a game, eliciting preferences among neighbouring games, and selecting which of them to move to based on those preferences. A trajectory has terminated in an attractor game when no neighbouring game is strongly preferred to the current one. Under this definition, two neighbouring games can both be attractors, permitting us to define a set of contiguous attractor games as a basin. Basins are neutral; players have no preferences between attractor games in a basin.

# 4. The self-interested dynamic

Given this framework, we first define a dynamic based on rational, self-interested agents who change the games they play with an eye to institutionalizing their profits and position. This generalizes the absolute fitness maximizing agent of economic game theory to the institutional evolutionary context. Here, artificially selfish agents converge on prosocial outcomes, indicating that more realistic agents are at least as likely to do the same.

## (a) Within-game behaviour

Within-game, agents in this self-interested dynamic play under a model of rationality, selecting unique pure-strategy Nash equilibria when they exist and mixed-strategy equilibria otherwise, randomizing over equilibria when several of one type exist.

## (b) Between-game behaviour

Across games, a player's institutional preferences define their trajectory towards an attractor. Agents in the self-interested dynamic prefer games that are stable, predictable and efficient; they prefer a game with a Nash equilibrium that is unique (stable), that is in pure strategies (predictable), and that includes the focal player's top-ranked outcome (efficient). This agent has no social preferences: given two games that are equally stable, predictable and efficient, players are indifferent as to whether one provides better or worse outcomes to another.

It is important to note here that seemingly familiar concepts such as stability, efficiency, predictability and fairness, are quite different as we define them. They are usually understood as

properties of game outcomes. We reintroduce them here as properties of whole games, imposing a distinction between outcome features and game features. For example, an outcome is traditionally seen as stable if it is a Nash equilibrium, while in this work a game is stable as well, if it has exactly one Nash equilibrium of a certain type. Agent preferences for game-level features are central to the between-game component of institutional evolutionary dynamics.

## (c) Aggregation rule

The self-interested dynamic's aggregation rule is a simple implementation of a rational agent working to consolidate a beneficial position. The player with greater earnings after the first randomly selected game becomes the focal player and chooses the next game in the trajectory. Any ties in pay-offs received at one step of the trajectory break in favour of the focal player who chose that game, with the ties in the first game breaking randomly. This approach to aggregation amplifies the selective strength of selfishness to create a setting that is inhospitable to socially beneficial outcomes. With these simple assumptions, we ensure that one player drives the whole trajectory in each simulation run. Thus, in an analogue to regulatory capture, power within a system confers power over it.

## 5. Measures

Generally, we are interested in the attractor games and how they differ from games in the broader space. Specifically, we are interested in how inequality properties change in attractors, a question that is especially interesting in the self-interested dynamic, whose agents do not prefer either equality or inequality. Establishing the effects of institutional evolution on emergent inequality is especially important given the theorized effects of inequality on institutional evolution. Political economic modelling of unequal policy influence—political power—shows that such deviations lead to lower quality governance, in the sense of less efficient management of public goods [47]. We offer two measures of equality. One is a space's proportion of 'win–win' games, games in which two players share the same top-ranked outcome. Another more sensitive and continuous measure of equality is the Gini coefficient of the pay-offs of each game's equilibrium outcomes.

Gini is a familiar nonparametric equality measure that is easily generalized to discrete pay-offs. For a two-choice $n$-player normal-form game, there are $2^n$ outcomes, each with $n$ pay-offs valued 1 to $2^n$ (the number of ranks is the number of outcomes to rank). Pay-offs within an outcome may be nearly equal to each other or widely varied, a property that Gini can determine. Under this measure, an equilibrium outcome that one player ranks highly and others rank poorly will receive a high Gini score close to the maximum value of 1, while an outcome in which all players receive the same pay-off (whether all high or all low) will be closer to the minimum value, 0.

## 6. Results

## (a) Two-player games with self-interested agents are disproportionately fair

Although this framework for institutional evolution can articulate many questions about institution-level change

processes, our motivating questions in this specific model concern the properties of the games selected by institutional evolution: the attractors. How many attractor institutions are there, how do they differ from the broader space, and how do the values and features they represent differ from the values of the agents that selected them?

Under the self-interested dynamic, a game is an attractor if it has a unique pure-strategy Nash equilibrium that pays the maximum pay-off to the focal player. The attractor games are a subset of the games with exactly one Nash equilibrium.

Contrary to intuition, 'win–win' games are very numerous in the two-player space: one in four games in the space are win–win (figure 3a). However, not all attractors are win–win and not all win–win games are attractors. For example, the game space includes a representation of the Stag Hunt, which is win–win by our definition but has a second equilibrium that sets it outside of the set of attractors, while the Samaritan's Dilemma is in the attractor set, despite not being win–win.

Shifting focus from the baseline properties of the space to the properties of trajectories through it, we used exhaustive numerical search to find that 54 out of 144 (37.5%) of the two-player games are attractors, and that they form a single contiguous basin (figure 3b). Under the preferences we defined, this basin is neutral in the sense that no game within it is preferable to any other. Of games in this basin, one-half are win–win, compared to one-quarter of all two-player games. This is our main result: the self-interested dynamic doubles the concentration of win–win games in the two-player space, despite the self-interested agent's absence of social preferences.

## (b) Attractors in the $n$-player games

Generalizing these results to $n$ players, we gain further insight into the effects of between-game preferences on institutional evolution. We find, based on the properties observed over large samples of randomly drawn games, that attractor games constitute a steadily decreasing proportion of all games as $n$ increases (figure 4a).

This might at first glance seem to imply that dynamics become more important in steering evolution towards certain types of attractor games, as the vanishing number of attractors and increasing number of games contribute to an increase in the average number of steps to convergence. However, attendant with the effects of population growth, driven mainly by the increasing dimensionality of constituent games, is an explosion in the number of nearest neighbours per game (at the rate reported above), which in turn drives an explosion in the number of shortest paths between arbitrary pairs of games, and, ultimately, very unconstrained dynamics.

## (c) Scaling of inequality in the self-interested dynamic

Focusing again on the self-interested dynamic, we look more closely at questions of equality. With the explosion in the number of games with increasing $n$ comes a crash in the proportion of win–win games (including 'win–win–…–win' games): $2^{-n^2}$.

This figure is based on the number of games in which one outcome contains the top-ranked pay-offs of all $n$ players, divided by the number of games with $n$ players. As a fraction of the total number of games, this value decreases super-exponentially from one in four two-player games being

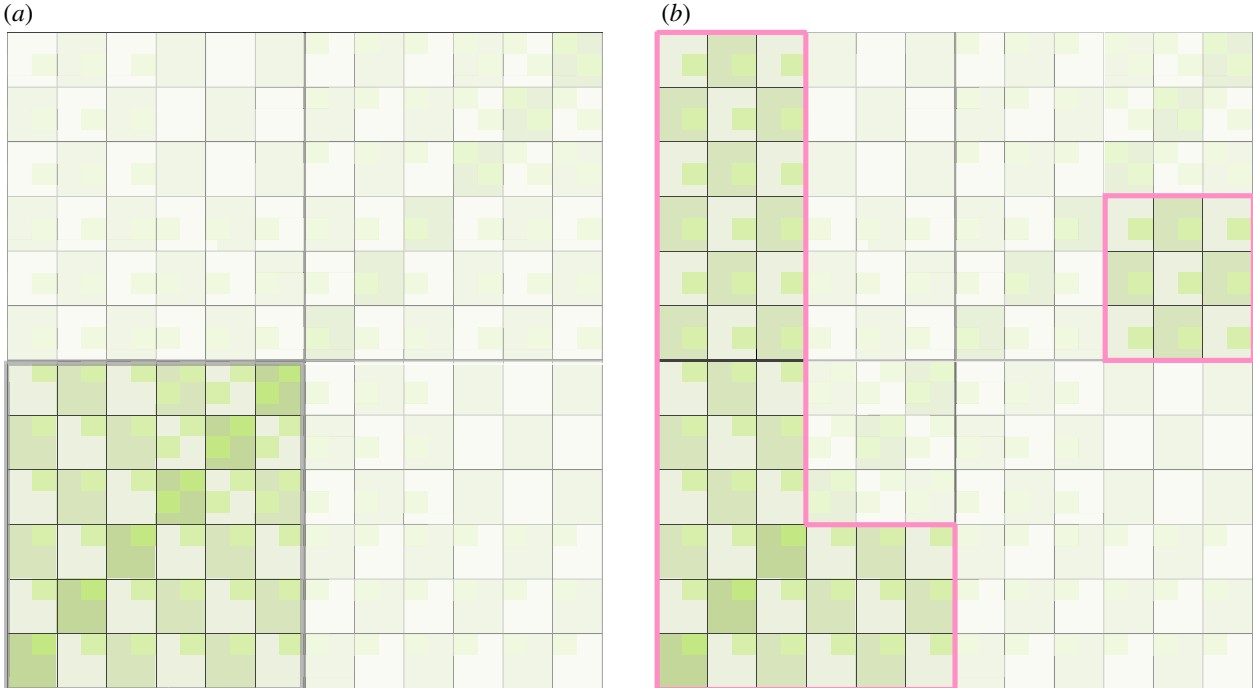

**Figure 3.** The win–win games and the attractor games of the institutional dynamic. (*a*) Win–win games account for one-quarter (36 out of 144) of all games in the two-player space, shown here in the lower-left quadrant. See figure 2*a* for the details of each game, namely that each has an outcome conferring the maximum pay-off of 4 to both players. (*b*) The pink outline shows the basin of attractors that results from self-interested agents' evolutionary trajectories. Note that these attractor games form a contiguous block; the nine games on the right have several neighbours among the games in the block on the left, via swaps that are not apparent from this two-dimensional representation of the space. In truth, each quadrant is a torus with links to the other tori. Note also that one-half (27 out of 54) of these attractor games are in the win–win quadrant. Compared to panel (*a*), the institutional evolutionary process doubles the chances that a randomly drawn game will be a win–win game, even though the selfish agents driving it have no explicit preferences for mutually beneficial games. (Online version in colour.)

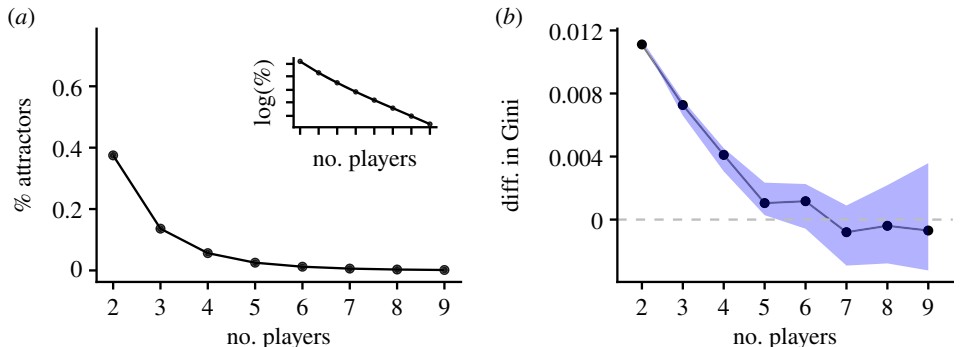

**Figure 4.** Emergent fairness driven by the self-interested dynamic becomes negligible as the player population grows. (*a*) Attractors become a smaller fraction of games as population (number of players) increases. The black line, derived from simulation, shows the computed proportion of games with up to nine players that are attractors of the self-interested dynamic. (*b*) Each game outcome contains pay-offs for each player, and players can receive very different pay-offs from the same outcome. We compute the Gini coefficient of the pay-offs in the Nash outcomes of games, comparing the outputs of all games to attractor games. We find that the difference quickly becomes negligible as population size increases, reinforcing the argument that the self-interested dynamic's incidental selection for equality disappears quickly as the population increases. (Online version in colour.)

win–win, to fewer than one in a billion win–win games at $n = 6$. As a fraction of attractors, which themselves constitute a declining proportion of large-$n$ games, attractor games that are also win–win become rare much more quickly, such that the win–win property becomes exceedingly rare even among agents that are selecting for them.

For a more sensitive measure, we also compare the Gini coefficients over the pay-offs of Nash outcomes of randomly drawn attractor and non-attractor games (figure 4*b*). The difference between them quickly becomes negligible, consistent with our finding that, in games with a large number of players, the

self-interested dynamic selects for games that are only desirable to the single favoured agent driving the dynamic.

## 7. Discussion

The interactions that structure our daily lives are not randomly selected from the space of social systems, nor from the small subset of games, such as the Prisoner's Dilemma and Stag Hunt, that have proved most useful for illustrating theoretical distinctions. Institutions and other social

structures—languages, rites, and systems of culture—develop through a process that can be conceptualized as a trajectory through institution space. When agents have preferences over games, and the ability to make incremental changes to those games, they can dramatically remake the space of expressed institutions. In particular, we find that agents who navigate a simple game space in a self-interested manner experience a dramatic increase in the proportion of win–win outcomes, from 25% to 50% of available games. Self-interested agents converge on a subset of games that are disproportionately fair in the equilibrium outcomes they provide. It appears that games which are predictable, stable, and efficient are also more likely to be incidentally fair. This particular property does not scale with game size, whether measured in terms of the number of players or the number of choices. Incidental fairness very quickly becomes negligible as populations of self-interested agents grow.

## (a) An alternative account of norms and conventions

One contribution of this work is to offer an alternative account of informal institutional constraints, such as norms, conventions, and other proposed mechanisms for the real-world state of affairs in a society. Existing game-theoretic conceptions of norms, conventions, and other proto-institutional structures understand these constructs as emergent regularities in within-game behaviour. But agents who are subject to a norm or convention do not just converge on one versus another pattern of behaviour, they experience different pay-offs, consequences, and new strategic affordances. In our simulations, a norm or convention emerges as a result of institutional dynamics that drive agents to transform strategically loaded social dilemmas into games in which their interests are naturally aligned (win–win) or orthogonal (no-conflict; see figure 1*b* for an example). Within the proposed institutional evolutionary framework, the convergence of a population of agents upon some stable pattern of socially efficient behaviour is largely a function of institution-scale processes, rather than strictly behavioural processes.

## (b) The pair as the most common scale of institutional organization

The advantages we find to small social systems, especially those of size two, may help explain the ubiquity of pairs as a core unit of social organization, and the challenges faced by corporate entities of all types as they become large. As social systems grow larger, they become susceptible to elite capture, unfairness and runaway concentrations of power [21,48,49]. In the other direction, persistent mating pairs (including marriages) are an organizing principle common to many human and other animal societies [50].

While theories of these general phenomena typically look to individual incentives and behaviour for mechanisms, we suggest that equality and inequality can be assured by the combinatorics of small- and large-$n$ game spaces, regardless of agent preferences.

## (c) A mechanism for major transitions in human evolution

Human evolutionary history has been punctuated by major transitions in which qualitative shifts to new forms of organization have fundamentally altered societies. Take, for example, the transition of human sociality from the relative fitness maximization of biology's evolutionary game theory, through the absolute fitness maximization of economic game theory, to the relative group-fitness maximization of multi-level selection theory. In the within-game framework of behavioural evolution, it is not clear how major transitions in social structure might fall out of changes in within-game outcomes. But under our model, the ability to change the institutional context, as well as small changes to the preferences driving that ability, amplify differences between otherwise subtle game-theoretic distinctions and provide a mechanism for major transitions in human evolution.

## (d) Extensions and limitations

The space we introduce in this work is artificial. For example, real social interactions are characterized by large, poorly defined choice sets that evolve over time and interact in complex ways. Information is limited and empirical violations can be found for virtually every theoretical assumption. Even setting aside real-world relevance, the role of simple $2 \times 2$ normal-form games in theory is increasingly in question, as methods evolve for incorporating ever more richness into game-theoretic models. Our results hold for a narrow subset of games, namely those two-choice normal-form games with ranked pay-offs. This makes it impossible to test other institutional preferences, such as those for extensive- versus normal-form games, repeated versus single-shot relationships, more versus fewer choices, or more versus less game complexity. Nevertheless, the present work is not unique for suffering from these shortcomings, which are typical of even the most impactful applications of game theory to social and behavioural sciences.

The applicability of our specific fairness finding is narrow: institutional evolution seems to only be a mechanism for emergent fairness in the case of a small number of agents facing a small number of choices. We suggest that it is the overall decrease in win–win games with game size that decreases their prevalence in attractors. Increasing either population or number of choices decreases the probability that a game will be win–win; that it will contain an outcome that provides maximum utility to *all* players. An increasing number of choices decreases this probability by increasing the number of outcomes to rank, the denominator, without increasing the number of top-ranked outcomes, the numerator (by definition: one). An increasing population also increases the number of outcomes to rank, while decreasing the probability that any of those outcomes will contain all top pay-offs.

The agents we introduce are artificial as well, but again, they are artificially hostile to cooperation. If the dynamic we describe drives the most self-interested agents to prosocial outcomes, agents with minimal pro-social biases are at least as likely to do the same.

Against this promising background, our institutional evolutionary framework makes behavioural studies simple to articulate. In one design we have developed, two participants play a randomly selected game from the space of games, and a game that neighbours it, and are then allowed to choose which of the two to play a second time. By repeating this procedure, one can directly compute the attractor games that those preferences drive dynamics towards.

By explicitly modelling game change dynamics, the proposed framework makes it possible to test the effects of

dynamical phenomena such as history dependence, neutral evolution, and the coevolution of within-game institutional experiences and between-game institutional preferences [51–53] as well as emergent diversity, heterogeneous agents and the interaction between rules and culture. For an example of possible extensions, consider the variety of aggregation rules. In the dynamic we consider here, the 'winner' of an initial game outcome gains unilateral control over all subsequent choices of game, ensuring that a first-time winner will continue to win. However, in simple variations the choice of game could be driven by the choice of a randomly selected agent, or the preferences of a majority or plurality of players, allowing us to incorporate increasingly complex forms of collective action into models of the institutional evolutionary process.

## 8. Conclusion

We advance a view that humans and other animals are not caged subjects of immutable institutions. Agents can change their incentives as those incentives change them. These institutional change processes are of fundamental interest to both evolutionary and behavioural game theories in general, and institutional analysis in particular. Still, we have been lacking tractable frameworks for representing the richness of institutional evolution. Dynamics over game spaces offer a parsimonious representation of institutional evolutionary processes. Within our framework, institutional change is a trajectory through neighbouring games, in which players evolve the games they play by incrementally making their pay-off structures more favourable. We find that the mere combinatorics of game spaces can impose constraints that incidentally encourage socially beneficial outcomes, at least in small social systems. Elucidating the properties of 'attractor' institutions sheds light on the emergence of organized human groups.

Data accessibility. Simulation code and data are available at: https://zenodo.org/record/4301086.

Authors' contributions. S.F. conceived of the research, computed the numerics and wrote the manuscript. C.A. conceived of the research and contributed to the manuscript.

Competing interests. We declare we have no competing interests.

Funding. This work was supported in part by the Neukom Institute for Computational Science.

Acknowledgements. The authors wish to thank Bryan Bruns, Austin Shapiro, Pete Richerson, Monique Borgerhoff Mulder, Cristina Moya, and the Evolution and Ecology of Human Behavior and Culture group at the University of California, Davis.

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
