## [Reviewer comments · Proceedings of the Royal Society B: Biological Sciences]

Review History

RSPB-2020-1050.R0 (Original submission)

Review form: Reviewer 1

Recommendation

Reject – article is scientifically unsound

Scientific importance: Is the manuscript an original and important contribution to its field?

Marginal

General interest: Is the paper of sufficient general interest?

Acceptable

Quality of the paper: Is the overall quality of the paper suitable?

Poor

Is the length of the paper justified?

No

Should the paper be seen by a specialist statistical reviewer?

No

Do you have any concerns about statistical analyses in this paper? If so, please specify them explicitly in your report.

No

It is a condition of publication that authors make their supporting data, code and materials available - either as supplementary material or hosted in an external repository. Please rate, if applicable, the supporting data on the following criteria.

Is it accessible?

No

Is it clear?

No

Is it adequate?

No

Do you have any ethical concerns with this paper?

No

Comments to the Author

The manuscript presents a new perspective in investigating evolutionary dynamics of human social institutions.

The Authors claim that they "... introduce a framework for modelling how agents change the games they are placed in. We contrast this between-game "institutional evolution" with the more familiar within-game "behavioural evolution". Although game theory is normally used to model social systems individually, trajectories through larger spaces of games can be helpful for simulating social change."

Nonetheless, the complex (adaptive and potentially, social-ecological) system human social institutions are an example of results oversimplified: the modelling approach suggested, based on 2-player social dilemma games, does not seem to be able to capture the complex nature of human social institutions. Many individuals act and interact in order to solve such social dilemmas often adopting sophisticated strategies and certainly formulating first and agreeing then on the rules of the games they play.

The well established theoretical framework of human social institutions conceived as series of constitutional / political / economic games is not even mentioned in the manuscript.

Finally, the quantitative analyses presented in support of this new innovative way to conceive human social institutions as trajectories in series of 2-players games are poor and not sufficiently detailed; moreover, results basically show that the whole modelling approach proposed works well only for 2-players games, which seems to be very restrictive and not taking into consideration the multi-player / multi-stage / multi-scale nature of human social institutions.

The lack in the availability of the code used to run the simulations and the analyses does not help in properly understanding the quality and the details of the proposed modelling approach.

Review form: Reviewer 2

Recommendation

Major revision is needed (please make suggestions in comments)

Scientific importance: Is the manuscript an original and important contribution to its field?

Excellent

General interest: Is the paper of sufficient general interest?

Good

Quality of the paper: Is the overall quality of the paper suitable?

Good

Is the length of the paper justified?

Yes

Should the paper be seen by a specialist statistical reviewer?

No

Do you have any concerns about statistical analyses in this paper? If so, please specify them explicitly in your report.

No

It is a condition of publication that authors make their supporting data, code and materials available - either as supplementary material or hosted in an external repository. Please rate, if applicable, the supporting data on the following criteria.

Is it accessible?

No

Is it clear?

N/A

Is it adequate?

N/A

Do you have any ethical concerns with this paper?

No

Comments to the Author

Evolution of human institutions is an underdeveloped area of great importance and significance. The authors utilize/develop an interesting and apparently powerful approach in which institutions evolve as a result of selfish agents choosing institutions that are stable, predictable, and efficient. They also show that such institutional evolution can lead to fairness. Overall I like their approach and the paper a lot.

I believe it can be rather stimulating for the field. My comments below are mostly about clarification of mathematical results and references to similar work.

- make it clear in the abstract that in contrast to the evolutionary game theory models describing populations of players (which are more familiar to the readers of this journal) here the games are played by the same small group of individuals.

- I'd suggest to frame your approach in terms of Hurwicz's "Institutions as families of game forms".

- discuss somewhat analogous work using different models of institutional evolution (e.f. by Bednar and Page).

- p.3, line 15: I am not sure what "double the baseline" means here.

- explain the origin of some mathematical results, e.g. 6/9-10; 6/16-18; 6/22, 10/6, 10/33

- 8/25: discuss mathematical models in economics in which agents with larger power get to set the rules of the economic game (e.g., Acemoglu, Robinson, Sonin, Bisin and Verdier, Gorodnichenko and Roland)

- 10/7-8: 1e3 ?? and 1e4000 ??

- mention/discuss Gavrilets 2012 paper on the evolutionary origins of the egalitarian behavior.

- mention/discuss Powers and Lehman's model of the evolution of institutions.

Decision letter (RSPB-2020-1050.R0)

03-Jun-2020

Dear Dr Frey:

I am writing to inform you that your manuscript RSPB-2020-1050 entitled "A dynamic over games drives selfish agents to win-win outcomes" has, in its current form, been rejected for publication in Proceedings B.

This action has been taken on the advice of referees, who have recommended that substantial revisions are necessary. With this in mind we would be happy to consider a resubmission, provided the comments of the referees are fully addressed. However please note that this is not a provisional acceptance.

Sincerely,
Dr Locke Rowe
mailto: proceedingsb@royalsociety.org

Associate Editor

Board Member: 1

Comments to Author:

The two reviewers agree that this manuscript offers a potentially interesting new perspective to understanding the evolution of social institutions, but they both raise a number of important concerns. Reviewer 1, in particular is rather critical, arguing that two-player games are insufficient to capture the complex, multi-scale nature of human social institutions. Of course any model is, by necessity a simplification of reality, but it is important to clarify the argument that the game theoretical approach can provide important insights in this context. This reviewer also raises concerns about the lack of detail regarding the quantitative analyses in the paper. Reviewer 2 is much more positive, but stresses the need for further explanation of the mathematical results and coverage of relevant literature. I would be happy to consider a revised version of the manuscript if you are able to address the reviewers' concerns. Finally, I note that the github link provided for the simulation code and data does not seem to be working (I get a "page not found" error). Please ensure that all the relevant data and code are available for reviewers to evaluate in the resubmission.

Reviewer(s)' Comments to Author:

Referee: 1

Comments to the Author(s)

The manuscript presents a new perspective in investigating evolutionary dynamics of human social institutions.

The Authors claim that they "... introduce a framework for modelling how agents change the games they are placed in. We contrast this between-game "institutional evolution" with the more familiar within-game "behavioural evolution". Although game theory is normally used to model social systems individually, trajectories through larger spaces of games can be helpful for simulating social change."

Nonetheless, the complex (adaptive and potentially, social-ecological) system human social institutions are an example of results oversimplified: the modelling approach suggested, based on 2-player social dilemma games, does not seem to be able to capture the complex nature of human social institutions. Many individuals act and interact in order to solve such social dilemmas often adopting sophisticated strategies and certainly formulating first and agreeing then on the rules of the games they play.

The well established theoretical framework of human social institutions conceived as series of constitutional / political / economic games is not even mentioned in the manuscript.

Finally, the quantitative analyses presented in support of this new innovative way to conceive human social institutions as trajectories in series of 2-players games are poor and not sufficiently detailed; moreover, results basically show that the whole modelling approach proposed works well only for 2-players games, which seems to be very restrictive and not taking into consideration the multi-player / multi-stage / multi-scale nature of human social institutions. The lack in the availability of the code used to run the simulations and the analyses does not help in properly understanding the quality and the details of the proposed modelling approach.

Referee: 2

Comments to the Author(s)

Evolution of human institutions is an underdeveloped area of great importance and significance. The authors utilize/develop an interesting and apparently powerful approach in which institutions evolve as a result of selfish agents choosing institutions that are stable, predictable, and efficient. They also show that such institutional evolution can lead to fairness. Overall I like their approach and the paper a lot.

I believe it can be rather stimulating for the field. My comments below are mostly about clarification of mathematical results and references to similar work.

- make it clear in the abstract that in contrast to the evolutionary game theory models describing populations of players (which are more familiar to the readers of this journal) here the games are played by the same small group of individuals.
- I'd suggest to frame your approach in terms of Hurwicz's "Institutions as families of game forms".
- discuss somewhat analogous work using different models of institutional evolution (e.f. by Bednar and Page).
- p.3, line 15: I am not sure what "double the baseline" means here.
- explain the origin of some mathematical results, e.g. 6/9-10; 6/16-18; 6/22, 10/6, 10/33
- 8/25: discuss mathematical models in economics in which agents with larger power get to set the rules of the economic game (e.g., Acemoglu, Robinson, Sonin, Bisin and Verdier, Gorodnichenko and Roland)
- 10/7-8: $1e3$?? and $1e4000$??
- mention/discuss Gavrilets 2012 paper on the evolutionary origins of the egalitarian behavior.
- mention/discuss Powers and Lehman's model of the evolution of institutions.

Author's Response to Decision Letter for (RSPB-2020-1050.R0)

See Appendix A.

RSPB-2020-2630.R0

Review form: Reviewer 1

Recommendation

Accept as is

Scientific importance: Is the manuscript an original and important contribution to its field?

Good

General interest: Is the paper of sufficient general interest?

Good

Quality of the paper: Is the overall quality of the paper suitable?

Good

Is the length of the paper justified?

Yes

Should the paper be seen by a specialist statistical reviewer?

No

Do you have any concerns about statistical analyses in this paper? If so, please specify them explicitly in your report.

No

It is a condition of publication that authors make their supporting data, code and materials available - either as supplementary material or hosted in an external repository. Please rate, if applicable, the supporting data on the following criteria.

Is it accessible?

Yes

Is it clear?

Yes

Is it adequate?

Yes

Do you have any ethical concerns with this paper?

No

Comments to the Author

Authors had addressed the comments made in the previous review and they had clarified some details of the suggested modelling approach, listing advantages and limits of the modelling framework.

Still, I am not 100% sure about the fact that the main results Authors show are not true for games with more than 2 players / 2 strategies. But, I see the advantage in adopting a simple and clear theoretical framework.

Review form: Reviewer 2

Recommendation

Accept as is

Scientific importance: Is the manuscript an original and important contribution to its field?

Excellent

General interest: Is the paper of sufficient general interest?

Good

Quality of the paper: Is the overall quality of the paper suitable?

Good

Is the length of the paper justified?

Yes

Should the paper be seen by a specialist statistical reviewer?

No

Do you have any concerns about statistical analyses in this paper? If so, please specify them explicitly in your report.

No

It is a condition of publication that authors make their supporting data, code and materials available - either as supplementary material or hosted in an external repository. Please rate, if applicable, the supporting data on the following criteria.

Is it accessible?

Yes

Is it clear?

Yes

Is it adequate?

Yes

Do you have any ethical concerns with this paper?

No

Comments to the Author

Thanks for the revision.

On p.1 please fix: "A more recent approach approaches institutions..."

Decision letter (RSPB-2020-2630.R0)

12-Nov-2020

Dear Dr Frey

I am pleased to inform you that your manuscript RSPB-2020-2630 entitled "A dynamic over games drives selfish agents to win-win outcomes" has been accepted for publication in Proceedings B.

The referee(s) have recommended publication, but also suggest some minor revisions to your manuscript. Therefore, I invite you to respond to the referee(s)' comments and revise your manuscript. Because the schedule for publication is very tight, it is a condition of publication that you submit the revised version of your manuscript within 7 days. If you do not think you will be able to meet this date please let us know.

[http://datadryad.org/submit?journalID=RSPB&manu=\(Document not available\)](http://datadryad.org/submit?journalID=RSPB&manu=(Document%20not%20available)) which will take you to your unique entry in the Dryad repository. If you have already submitted your data to dryad you can make any necessary revisions to your dataset by following the above link. Please see <https://royalsociety.org/journals/ethics-policies/data-sharing-mining/> for more details.

Sincerely,
Dr Locke Rowe
mailto: proceedingsb@royalsociety.org

Associate Editor

Comments to Author:

Thank you for your thorough efforts in revising the manuscript. The reviewers found it much improved and I am happy to accept it for publication, pending two very minor edits: (1) R1 points out some awkward phrasing in the introduction ("a more recent approach approaches") and (2) It may be worth including a sentence in the Discussion to address R2's query: "I am not 100% sure about the fact that the main results Authors show are not true for games with more than 2 players / 2 strategies" (note this latter point could be useful but I do not see it as essential".

Reviewer(s)' Comments to Author:

Referee: 2

Comments to the Author(s).

Thanks for the revision.

On p.1 please fix: "A more recent approach approaches institutions..."

Referee: 1

Comments to the Author(s).

Authors had addressed the comments made in the previous review and they had clarified some details of the suggested modelling approach, listing advantages and limits of the modelling framework.

Still, I am not 100% sure about the fact that the main results Authors show are not true for games with more than 2 players / 2 strategies. But, I see the advantage in adopting a simple and clear theoretical framework.

Decision letter (RSPB-2020-2630.R1)

19-Nov-2020

Dear Dr Frey

I am pleased to inform you that your manuscript entitled "A dynamic over games drives selfish agents to win-win outcomes" has been accepted for publication in Proceedings B.

Your article has been estimated as being 9 pages long. Our Production Office will be able to confirm the exact length at proof stage.

Open Access

Paper charges

Sincerely,

Appendix A

Response to reviewers: A dynamic over games drives selfish agents to win-win outcomes

Associate Editor

Board Member: 1

Comments to Author:

The two reviewers agree that this manuscript offers a potentially interesting new perspective to understanding the evolution of social institutions, but they both raise a number of important concerns. Reviewer 1, in particular is rather critical, arguing that two-player games are insufficient to capture the complex, multi-scale nature of human social institutions. Of course any model is, by necessity a simplification of reality, but it is important to clarify the argument that the game theoretical approach can provide important insights in this context. This reviewer also raises concerns about the lack of detail regarding the quantitative analyses in the paper. Reviewer 2 is much more positive, but stresses the need for further explanation of the mathematical results and coverage of relevant literature. I would be happy to consider a revised version of the manuscript if you are able to address the reviewers' concerns.

We are gratified that both reviewers see the potential interest of this work. In this resubmission we have endeavored to clarify that, despite its artificial simplicity, the game theoretic approach is appropriate given the goals of this work. We have improved the detail and clarity of the formal and quantitative results, reducing and reformulating them to focus less on background and more on our contribution: the selection of unexpectedly fair games despite strictly self-interested agents. And we have made more fair and meaningful connections to relevant prior work. We also trimmed the paper while making room for suggestions. The revised paper is slightly shorter, including new references and clarifications.

Reviewer(s)' Comments to Author:

Referee: 1

Comments to the Author(s)

The manuscript presents a new perspective in investigating evolutionary dynamics of human social institutions.

The Authors claim that they "... introduce a framework for modelling how agents change the games they are placed in. We contrast this between-game "institutional evolution" with the more familiar within-game "behavioural evolution". Although game theory is normally used to model social systems individually, trajectories through larger spaces of games can be helpful for simulating social change."

Nonetheless, the complex (adaptive and potentially, social-ecological) system human social institutions are an example of results oversimplified: the modelling approach suggested, based on 2-player social dilemma games, does not seem to be able to capture the complex nature of human social institutions. Many individuals act and interact in order to solve such social dilemmas often adopting sophisticated strategies and certainly formulating first and agreeing then on the rules of the games they play.

Focusing on the over-simple 2x2 games, we report findings of interest to the researchers in many disciplines who have found value in those games over the decades despite their shortcomings. The case for seeing our model as a contribution despite its simplicity is that we take this well-trod domain — one that a reasonable person could say has long been wrung dry of big insights — and approach it from a direction that nevertheless illuminates questions on the timely topic of cultural evolution.

Our toy model most definitely “oversimplifies the adaptive and potentially social-ecological system that human social systems are an example of”. We manage this potential overreach in two ways, 1) by distinguishing between our model (which does not capture the suggested additional complexity) and our general modeling framework (which can), and 2) by presenting not just any artificial over-simple model, but a model that is artificially over-simplistically hostile to cooperative outcomes. Our argument is that if cooperation or fairness emerge anyway, then the mechanism we explore is strong enough to be general in more realistic spaces, or at least a contribution worth reporting.

General modeling framework

In the section Institutional Evolution, we endeavored to distinguish between our descriptions of the general institutional preferences framework and our own specific instantiation of it. As we describe in the paper, to work under this framework requires defining five parts: a space of possible institutions, a definition of distance between institutions, models of agent behavior within and also between institutions, and an aggregation rule for turning each individual’s institutional preference into the collective’s choice.

After introducing the general framework, we use one bare-bones specification of these parts to define and present as our main result an instantiation of the framework that yields original insights. To make the distinction between model and modeling framework clear, here is a brief representation of the elements of our specification of the framework:

- Institution space = *2x2 games*
- Distance = *transposition distance*
- Agent, within-game = *Nash*
- Agent, between-game = *Nash-like self-interest*
- Aggregation = *Top-earning player’s preference prevails*

Although that specific implementation fits the description of the reviewers’ list of shortcomings, the framework itself is quite powerful enough to illuminate institutional dynamics in fully adaptive social-ecological systems like human social systems. At the current state of the art, it is not clear how to formally define an institution-space of that richness, but as the framework matures, nothing precludes growth in that direction.

For a more modest step in the direction of increasing complexity, we could, as the reviewer suggests, define agents who are capable of “*formulating first and agreeing then on the rules of the games they play.*” Under our institutional evolution framework, this suggestion would correspond to a change in aggregation rule, keeping all other components constant. To implement it, we would take the paper’s current Machiavellian “winner-take-all” aggregation rule, in which the highest earning player within a game selects the next game to maintain their

advantage, and replace it with any of a number of more cooperative or consensual mechanisms for combining all players' preferences into a collective outcome. For one example, simple voting or elections would be trivial to substitute in. For another, in a "live" laboratory implementation of our institutional evolutionary dynamic, with human instead of simulated agents, one could go even further and permit full-bandwidth deliberation between agents about which neighboring game to agree upon.

We have edited the manuscript to make these points more clear.

Conservative modeling choices

We maintain that any substantive deviation from our model in the direction of realism would make it more likely to yield cooperation, and therefore, that the mechanism we propose is likely to generalize to more realistic systems as well.

Therefore, while we are interested in more cooperative aggregation rules in general, they are not appropriate to the purposes of the submitted work. The rhetorical strategy of this work is to define an environment in which emergent cooperation is unrealistically unlikely, and show, under our mechanism, that it emerges anyway.

With the aggregation rule we chose, each trajectory is driven entirely by a single selfish agent who is choosing games entirely to maximize his own chances of retaining control and maximizing personal profits. Our result is that cooperative outcomes emerge in spite of the unfavorable conditions. Hopefully this explains why we selected such an artificially uncooperative approach. We are certainly interested in devoting future work to a thorough analysis of more cooperative (not to mention realistic) specifications of the framework.

We have edited the manuscript to make these points more clear.

Two-player restriction

There is one exception to our finding of robust emergent cooperation. One manipulation that we introduced seems to very effectively suppress emergent cooperative outcomes in our model. It speaks to this part of the reviewer's response:

... based on 2-player social dilemma games, does not seem to be able to capture the complex nature of human social institutions. ...

and later

... Many individuals act and interact in order to solve such social dilemmas ...

Although our main result, and most of our figures, focus on the 2x2 space, one of our main results, illustrated in Figure 4, is that we repeat our simulations up to 2x2x...x2 9-player games, to show that the mechanism we report seems only to work at the smallest scales of social organization. We highlight this result here to demonstrate that we are aware that many individuals act and interact to solve social dilemmas and change the rules of the game. Our simulations show that the ability of the model to encourage win-win outcomes declines sharply as the number of players increases.

We're apprehensive to belabor the point because in other comments the reviewer is clearly aware that we extend our results beyond two players. We desire to be clear about how our results already speak to some of the reviewers' concerns. That said, another concern to the same point is equally justified: we base our results on 2-choice "social dilemma games which do not seem to be able to capture the complex nature of human social institutions." The reviewer hints at this concern with arguments that:

Many individuals act and interact in order to solve such social dilemmas often adopting sophisticated strategies and certainly formulating first and agreeing then on the rules of the games they play. ...

and later

... not taking into consideration the multi-player / multi-stage / multi-scale nature of human social institutions. ...

A point we try to make in the manuscript, related to the larger point about conservative modeling, is that most such extensions can only make our results stronger, on the condition that they increase the proportion of non-win-win games to win-win games. In our reported results (≥ 2 players) and beyond them (> 2 choices, > 1 stage), it is certainly true that the combinatorics dramatically increase the proportion of non-win-win to win-win games, for the same reason that only an infinitesimal fraction of all possible long DNA sequences code viable organisms: very specific conditions must hold for a game to be win-win, and the number of ways to violate those conditions increases with number of players and number of choices. To lean on the Kareninism: "all win-win games are alike, while each non-win-win game is non-win-win in its own way."

This is not to say that changes to other components, such as the aggregation rule or agent behavior rules, couldn't forestall those dynamics.

We have worked to be careful to properly hedge any claims to generality, and have given those claims another pass in the submitted revision.

The well established theoretical framework of human social institutions conceived as series of constitutional / political / economic games is not even mentioned in the manuscript.

We originally took this legacy too much for granted. The reviewer is right that drawing more explicitly from it strengthens the manuscript.

Finally, the quantitative analyses presented in support of this new innovative way to conceive human social institutions as trajectories in series of 2-players games are poor and not sufficiently detailed;

We are open to the possibility that our approach to the analysis is poor. To address this concern, we have improved the detail and clarity of our explanation of the analysis, and implemented the other reviewer's advice on the specifics of how we conducted each component of the analysis.

We explained the intuition of all equations in plain language. We included more signposting about which quantities were numerical and based on sampling, and which were formal and computed exactly.

... moreover, results basically show that the whole modelling approach proposed works well only for 2-players games, which seems to be very restrictive

Allow us to distinguish between the modelling approach/framework, a model derived from that approach, and a result derived from that model. The reviewer seems to be saying that our model or modeling approach have failed because our result documents cooperation failure in larger groups. Our position is the opposite: our model is a success because it finds cooperation in the smallest groups and not larger groups. Indeed, that's one of our results. This kind of result has empirical support, and mechanisms for the difficulty of scaling cooperation have been proposed, but the right one is in dispute, and we offer a novel one. Where a more familiar Dunbar-style (cognitive limits) mechanism might say that cooperation fails in larger groups because humans lose the ability to keep track of all interactions, we offer a more fundamental mechanism, that failure falls out of the combinatorics of spaces of possible institutions.

More importantly, we offer a novel mechanism for cooperation success in very small groups, one that suggests that there may be something special about groups of size two, which seem able to evolve into mutually beneficial institutions even without prosociality. The mechanism seems to work because the group-beneficial game feature "win-winness" cooccurs incidentally with the self-beneficial game features like that selfish players seek out.

The lack in the availability of the code used to run the simulations and the analyses does not help in properly understanding the quality and the details of the proposed modelling approach.

We are embarrassed to report that the URL we submitted had a typographical error. We have fixed the URL and improved the quality and documentation of the posted code.

Referee: 2

Comments to the Author(s)

Evolution of human institutions is an underdeveloped area of great importance and significance. The authors utilize/develop an interesting and apparently powerful approach in which institutions evolve as a result of selfish agents choosing institutions that are stable, predictable, and efficient. They also show that such institutional evolution can lead to fairness. Overall I like their approach and the paper a lot. I believe it can be rather stimulating for the field. My comments below are mostly about clarification of mathematical results and references to similar work.

- make it clear in the abstract that in contrast to the evolutionary game theory

models describing populations of players (which are more familiar to the readers of this journal) here the games are played by the same small group of individuals.

It is helpful to get some insight into what barriers a new reader had to get over to see our approach in the way that we are trying to report it. With the reviewers' suggestions, we believe the manuscript is better able to communicate our approach to readers.

- I'd suggest to frame your approach in terms of Hurwicz's "Institutions as families of game forms".

- discuss somewhat analogous work using different models of institutional evolution (e.f. by Bednar and Page).

...

- mention/discuss Gavrilets 2012 paper on the evolutionary origins of the egalitarian behavior.

- mention/discuss Powers and Lehman's model of the evolution of institutions.

We have reworked the framing and discussion to involve more of the potentially relevant literatures. In particular, we took the opportunity to better understand the relationship of our work to Hurwicz's early contribution. Properly extended to encode the constraints of our topology, and the stopping rules for attractors, the most broad articulation of his approach is likely general enough to capture the process we describe, despite the fact that it was used to model a more top-down institutional change process in which architects impose a broad plan that they fill in iteratively with more detail.

Where a suggested literature or work was not relevant, we have used the suggested references as opportunities to highlight the key distinctions between familiar past work and our own.

- p.3, line 15: I am not sure what "double the baseline" means here.

It means that the effect size of our "treatment" is twice that of "control". In a baseline cultural evolutionary process that defines games as attractors randomly, the probability of converging upon a win-win game is 25%, while a process with self-interested institutional preferences converges up win-win games with 50% probability. We have clarified this point in the text.

- explain the origin of some mathematical results, e.g. 6/9-10; 6/16-18; 6/22, 10/6, 10/33

We have added more explanatory context to each equation.

- 8/25: discuss mathematical models in economics in which agents with larger power get to set the rules of the economic game (e.g., Acemoglu, Robinson, Sonin, Bisin and Verdier, Gorodnichenko and Roland)

Done.

- 10/7-8: 1e3 ?? and 1e4000 ??

10^3 and 10^{4000} . We have converted the text to this much more clear, concise, and familiar notation.